# Evaluating the Practices and Challenges of Youth Volleyball Development in Amhara Regional State, Ethiopia by Using the CIPP Model

**DOI:** 10.3390/healthcare10040719

**Published:** 2022-04-13

**Authors:** Zemenu Teshome, Bezabeh Wolde, Teketel Abrham, Tefera Tadesse

**Affiliations:** 1Sport Academy, Bahir Dar University, Bahir Dar P.O. Box 3001, Ethiopia; zemenutesh@yahoo.com (Z.T.); tekeselam@gmail.com (T.A.); 2Sport Science Department, Addis Ababa University, Addis Ababa P.O. Box 1176, Ethiopia; bzbhwold32@yahoo.com; 3Institute of Educational Research (IER), Addis Ababa University, Addis Ababa P.O. Box 1176, Ethiopia; 4Institute for Medical Education, University Hospital, LMU Munich, 80336 Munich, Germany

**Keywords:** CIPP model, program quality, youth volleyball, Ethiopia

## Abstract

Youth athletes’ talent identification and development has become a serious concern around the globe. However, empirical evidence regarding youth sports policies and practices is limited. Emphasizing the talent development needs of youth volleyball players, in this study, the authors evaluated the practices and challenges of a youth volleyball development program in Amhara Regional State, Ethiopia. This study addressed this concern by drawing upon Stufflebeam’s context, input, process, product (CIPP) model to explore a youth volleyball development program across the training sites located in Amhara Regional State, Ethiopia. With the help of this model, this study evaluated the prevailing contexts, allocated inputs, program implementation processes, and products. To this end, this study used multiple case studies involving ten youth project sites. The study participants included samples of participants (n = 179), consisting of youth volleyball players (n = 167), their coaches (n = 8), regional volleyball administrators, and regional educational office physical education coordinators (n = 4). The study participants identified some benefits from participating in the youth volleyball program, which included increased physical activity and health, enhanced positive interpersonal relationships, and knowledge of how to cope with challenges. However, they reported several challenges attributable to contextual constraints which included a lack of the necessary facilities and resources, lack of concern and convenient settings, poor implementation practices, and minimal outcomes. The findings suggest that the challenges of youth volleyball development in Ethiopia are complex and emanate from the context, input, process, and products. Accordingly, when addressing the issues of youth volleyball, it is necessary to develop systems, processes, methods, and tools that recognize all these concerns.

## 1. Introduction

Youth sports programs include those organized sports offered in many schools, clubs, and communities [1]. These sports programs are delivered to achieve multiple objectives, such as talent development, physical activity participation, and personal growth [2]. The benefits of participation in youth sports programs are primarily concerned with the facilitation of positive youth development outcomes as opposed to outright success as a team [3]. Consequently, there are strong theoretical and empirical links between sports coaching and athlete development. For example, transformational leadership behaviors have been theoretically linked to positive developmental outcomes within a youth sports context, while the coach–athlete relationship is a key tool used by coaches who aim to teach life skills to young athletes [4]. Thus, the theoretical focus is on the developmental skills gained from participation in youth sport. It is commonly believed that through sports, adolescents and youth learn important values and skills that will serve them well as they prepare for the rest of their lives [5]. Unfortunately, many youth sports development programs are neither well-structured nor implemented in a manner that these life skills can be learned in the sport and later transferred to other life domains [6]. 

A youth volleyball program focuses on developing all-around skills and abilities while helping players to become good team members with a passion for the game [7]. Young volleyball players also learn to share, take turns, and show empathy toward others [8]. If the youth volleyball program focuses on developing young players with skills in all aspects of the game and instills in them a drive to compete, they will develop team skills and a collective commitment to the success of the team. 

Previous research examined the personal factors of success for youth volleyball players, focusing on a quantitative analysis of early sports activities and their demographic, anthropometric, and physiological responses. However, few scholars have studied youth volleyball participation and identified conditions under which particular outcomes are likely to occur. Hence, there is a need for a conceptual framework and empirical evidence that identifies the contexts, inputs, and processes through which youth volleyball participation produces positive outcomes [9,10].

In Ethiopia, minimal research has been directed toward addressing aspects of youth volleyball from a broader perspective, including its policies and practices [11]. So far, only a limited number of studies have conducted a comprehensive assessment of youth sports programs [12,13], and not enough evidence is yet available to determine whether these programs are meeting their intended purposes. It is therefore crucial to holistically assess the volleyball program that is found in Amhara region. The purpose of this study was to investigate the practices and challenges in the development of the youth volleyball program in Amhara region. More specifically, this study answered the following basic research questions: What are the contextual factors in the program environment that are likely to influence the success of youth volleyball development in Amhara Regional State, Ethiopia?How is the program developed and practiced within the existing local and national contexts?What is the current practice of youth volleyball development across the different project sites or centers in Amhara Regional State, Ethiopia?What are the major challenges of the youth volleyball development program in Amhara Regional State, Ethiopia?

## 2. Materials and Methods

### 2.1. Study Design

This study used multiple case studies [14], involving ten cases (youth development project sites) to explore multiple program contexts found in Amhara Regional State. Given the dearth of literature documenting the quality of youth volleyball programs, utilizing multiple case studies is a useful exercise to provide rich and varied data. Multiple case studies allow a wider investigation of this study’s research questions. As this study is more intensely grounded in different empirical evidence, this type of case study also creates a more convincing evidence base. We used multiple case studies to understand the differences and similarities between the cases [15]. Another advantage is that the researchers can analyze the data both within each situation and across situations [16].

This study used a qualitative study approach because it helps to understand whether the studied youth volleyball program provides a climate that promotes sport-based youth development. In terms of framework, the CIPP evaluation model was used to holistically evaluate the youth volleyball development program. The practice of using a CIPP evaluation model as a framework for the study of youth sports program is similar to the practice of other studies conducted in sports sciences and sports management studies [17,18,19,20,21].

### 2.2. The Study Setting

There are 40 volleyball project sites in Amhara Regional State, which have been delivering youth volleyball trainings to both males and females. The development of volleyball programs is a process in which volleyball players are transferred, hired, or join the volleyball programs run by different sports academies and volleyball clubs based on their performance at the end of the season in which they engaged in the competition. Through this process, the most skilful or talented volleyball players in each volleyball program are selected and promoted or joined to the sports academy and hired from different volleyball clubs in developed countries. However, in the context of our country, Ethiopia, the concerned bodies and the public do not give special attention like this to other sports disciplines.

### 2.3. Conceptual Framework of the Study

The researchers selected the CIPP Model as a comprehensive framework that can capture the entire program components from the end users and the program owner’s perspectives [22]. The model’s core concepts are denoted by the acronym CIPP, which stands for evaluation of an entity’s context, input, processes, and products. Figure 1 presents the model components within the youth sports local and national contexts.

As shown in Figure 1, each element of the model addresses specific aspects of a youth volleyball development program. The first component of the model is the context that deals with the internal and external environment of the program. The second component is input, dealing with the resources needed to conduct the program. The third component is the process that describes how a youth development program is implemented. The fourth component is a product that describes the effectiveness and impacts of the program. 

### 2.4. Sample Participants Selection

The study participants included samples of participants (n = 179), comprising youth volleyball coaches (n = 8, males with more than five years of experience in youth coaching), youth volleyball players (n = 167), and Amhara region volleyball administrators. The administrators (n = 4, males involved in the administration of the youth volleyball program) included federation committees (n = 1), youth sports office experts in Amhara Regional State (n = 1), and zonal volleyball committee members (n = 1), and the regional education expert assigned for the management of youth sports program (n = 1). While the administrators were at the regional and zonal levels, office experts, and coaches were selected purposively depending on their roles in the programs. The youth volleyball players were selected by using a simple random sampling method. 

### 2.5. Instruments for Data Collection

Following the Stufflebeam [23] CIPP model, this study gathered qualitative data on the context, input, process, and product. The sources of data included youth volleyball players based on an open-ended questionnaire administered to 167 youth players from the 10-sample youth volleyball programs found in Amhara Regional State. Semi-structured interviews and FGD points were prepared for administrators and coaches, respectively. The authors also used document analysis. The open-ended items, the interview questions, and the FGD points are found in a “Appendix A” for this article.

#### 2.5.1. Documents as Sources of Qualitative Data

In this study, the authors used document analysis as one of the primary tools of data collection in which documents are interpreted by the authors to provide initial evidence-base information about youth volleyball programs in Amhara Regional State, Ethiopia. The types of documents used included public records, comprising official, ongoing records of the National Sports Commission, and its corresponding Amhara Regional State sports office activities. 

#### 2.5.2. Open-Ended Items in a Questionnaire

In the young volleyball players’ questionnaire, we included two open-ended question items that allow young participants to answer in open-text format so that they responded to the questions based on their complete knowledge, feeling, and understanding of the youth volleyball program. It means that the responses are not limited to a set of options. 

#### 2.5.3. Semi-Structured Interview

The purpose of the interview, in this qualitative inquiry, was to create a conversation that invites the key informants to tell stories, accounts, reports, and descriptions about their perspectives, insights, experiences, feelings, and emotions about the research questions [24]. In each semi-structured interview, the first author used a pre-planned interview guide to ask the interviewee relatively focused but open-ended questions about specific topics of interest related to youth volleyball program in Amhara Regional State.

#### 2.5.4. Focus Group Discussion (FGD)

In this study, FGD was used with youth volleyball coaches who have rich experience in coaching young athletes. The purpose of the FGD, in this qualitative inquiry, was to gain an in-depth understanding of the youth volleyball program in Amhara Regional State based on the views and perspectives of the youth coaches’ participants. In the focus group, we prepared semi-structured discussion points to enable a focused discussion among the youth coaches’ participants. In terms of the number of FGD participants, we involved eight youth coach participants, and this was acceptable as most focus groups in sports and exercise sciences research contain between four and ten participants [25]. The FGD points were particularly valuable sources of knowledge about the youth coaches’ experiences and the meaning attached to the volleyball youth program. 

#### 2.5.5. Observation Field Notes

The first author prepared field notes following each field observation. The field notes were qualitative notes recorded by the first author during or after his observation of a specific project site he was studying. The notes are intended to be read as evidence that gives meaning and helps in the understanding of the overall program feature found in each site as a specific case. Each field note describes the physical setting, the social environment, and the way in which youth volleyball players and their coaches interacted, and the participants and their roles in the setting.

### 2.6. Study Procedures

The first author was involved in a one-on-one interview with coaches. He also managed the FGDs with youth volleyball players. Each FGD consisted of 10 youth player participants. The data collection was handled by the first author in each sampled training site and he analyzed and interpreted the results by using thematic analysis. 

The authors used the federal and regional state program manuals as information sources about the program structure and implementation plans. These included the number of participants in each project site and the overall training duration, implementation, and follow-ups. Also, we used the Ethiopian national sports policy. These documents served as the source of evidence to evaluate the context and input related to the volleyball program studied.

To explore the setting and the coaching climate of the youth volleyball program studied, direct observations of training sessions and the entire sporting facilities were conducted (n = 10). The principal author was present at each project site to take observational field notes. The observer aimed to keep an open mindset during observation. Hence observations were unstructured, with field notes written down to document as much information as possible. The field notes included information about the project sites and how the coaching was organized and implemented.

Also, individual face-to-face interviews were used with the program administrators to explore their perception of the training context and the surrounding factors associated with the implementation of the program. Each interview lasted between 20 and 30 min. All interviews were held in person by the first author in a private room in the program setting. Due to this experience, a self-reflective stance was adopted during interviewing, with a general openness and curiosity about the interviewees’ experiences was established. Interview topics covered different aspects of the evaluated program, including the perceptions of the interviewee on the training session, the resulting effectiveness and impacts of the program, and the facilitating and hindering contextual factors. In addition, the interviewees reflected on the program implementation mechanisms and processes. 

Moreover, a focus group was created comprising the coaches involved in the program, lasting for about 90 min, with an audio recording. The first author facilitated the discussion. The study participants discussed their perceptions regarding the youth volleyball program, it’s supposed effectiveness and impact, facilitating or hindering contextual factors, and suggestions for a better future.

Ethical approval was provided by the Bahir Dar University Sports Academy Research Committee, and youth volleyball players and their parents provided written informed consent before participation. Each study participant was approached after gaining permission from the gatekeeper at the research site.

### 2.7. The Qualitative Data Analysis Methods

In this study, the authors used thematic analysis that finds themes in the text by analyzing the meaning of words and phrases [26]. We used both deductive and inductive coding at the different stages of the coding process. We used deductive coding at the beginning of the coding process guided by the conceptual framework. We started the deductive coding with a predefined set of four codes (Figure 2) and then assigned those codes to the qualitative data collected for this study. 

Then the sub-themes in each theme were coded by using inductive coding, also known as open coding. The inductive coding started from scratch, and we created codes based on the qualitative data itself. In our analysis, all the sub-themes and their attributes arise directly from the study participants’ responses. The inductive coding was an iterative process, which means it was thorough and relevant to the production of a more complete, unbiased look at the sub-themes throughout our data [27]. The inductive coding process included the following two steps. First, we coded each transcript line by line. As categories emerged, we used a constant comparative method to compare and refine categories as needed. 

We took all the data from the research dataset and categorized the themes. We organized the themes or codes based on the conceptual framework and how an effective youth development program may be understood from the structural, implementation, and outcomes perspectives [28,29].

This study used a qualitative dataset with a focus on exploring the context, input, process, and product, and the challenges surrounding the development and implementation of the youth volleyball program in the Amhara Regional State, Ethiopia. The focus here is to present the views and perspectives of youth athletes, their coaches, and the program administrators in their own words within the specific program contexts across different locations. For this, the authors took a qualitative research stance and the CIPP evaluation model to the study of youth volleyball program. The CIPP model suggests a specific approach to a youth sports program evaluation, comprising context, input, process, and product [22]. 

## 3. Results

Guided by the conceptual model, the results of the study are organized into four major themes, and each theme has 2–4 sub-themes that emerged from the empirical data collected for this study. Figure 2 presents an organizing framework of hierarchical themes.

The visual map presented in Figure 2 illustrates a hierarchical coding frame consisting of two levels as described below. 

The first-level code describes the major themes that the evaluation of this study focused on investigating (evaluating the context, input, process, and product).The second-level code specifies the sub-theme under each major theme. The attributes or features of each sub-theme are described in detail under each sub-theme.

This hierarchical framing supports the organization of codes based on the conceptual framework. It also allowed us to be able to generate a level of detail in a set of data.

### 3.1. Context Evaluation Results

In this study, context evaluation was used to give a rational reason for the youth volleyball program. This evaluation was conducted based on evidence in the following areas: policy needs and interests, institutional arrangements, talent development environment, and partnership. 

#### 3.1.1. Sports Policy: Needs and Interests

The Ethiopian national sports development policy declares the need to increase the pool of excellent athletes, thereby expanding the country’s potential to meet its aspiring target to excel in international sporting events [30]. In addition, according to the memorandum of understanding document signed by the National Sport Commission of Ethiopia and the Amhara Regional State Sport Commission office, there has been a strong mutual interest to scale up youth sports participation in Amhara Regional State. Based on this, there are 13 youth sports programs found in Amhara Regional State, in addition to the previous 3 youth and training programs [31]. 

According to the memorandum of understanding document, the youth volleyball program, as well as others, was designed to contribute to the ongoing sports development of the country by creating a healthy and dynamic youth sports system. The youth volleyball development program was targeted to promote youth volleyball players into senior volleyball players. Accordingly, 30% of the youth volleyball players who completed the age-appropriate training with better performance will go to the national sports academies and the other 68% will be transferred to various clubs [31]. In addition, according to the views of most of the coach FGD participants, both the accountability system as well as the coaches’ instructional approaches for the youth volleyball program in Amhara Regional State were not found effective as they had minimal professional development opportunities [32]. 

#### 3.1.2. Institutional Arrangements and Plans

The central concern of any youth sports program is the practice of an effective talent development program as planned in the program document. In this regard, the reviewed documents indicated that the volleyball talent development program consisted of 10 months of youth volleyball training, lasting from September to June. This range of months corresponds with the school activities in one academic year. Table 1 presents the days and hours distributions.

As indicated in Table 1, on average, the overall planned training lasts for 120 days, comprising a total of 240 h of training per year. However, the actual reality, as the first author verified in his field visit as well as in the evidence collected from the sampled coaches and youth players indicated, proper implementation, follow-up, and support did not appear as intended. In fact, clear follow-up and support structures were in place in the program planning documents; however this did not resonate in practice. For example, according to the youth coach participants response, program monitoring and evaluation, and provision of timely feedback for improvement were entirely absent. 

#### 3.1.3. Talent Development Environment

Many of the study participants, particularly youth volleyball players and their coaches, felt that the federal government and the regional state did not give equal attention to volleyball when compared with football and athletics. Most of the study participants perceived that the regional government did not focus on the overall development of volleyball for young people and for the success of the program. Based on this, they recommended that the National and Regional Volleyball Federation and the media should endorse a strong support for the development of youth volleyball. In particular, they should create a comfortable environment, providing the required support, and continually engage in monitoring and evaluating the program.

#### 3.1.4. Partnership with MoE and High Schools

The problems of volleyball programs in Amhara Regional State, as per the perspectives of different study participant groups, primarily included problems of ownership, coordination, support, and follow-ups. As one of the interviewed officials said, “the owner of the program is the school office, but 90% is owned by the sports commission.” One of the interviewees noted that switching the ownership from the sports commission to the office of education brings several advantages. 

### 3.2. Input Evaluation

#### 3.2.1. Sports Facilities, Equipment, and Materials

To effectively implement the youth volleyball development program, youth participants must have access to a court, a net, several volleyballs, and a pair of poles for attaching the net to the playing surface. Sporting resources, such as sporting suits, training facilities and equipment, as well as essential first aid kits, are key resources both for training and competitions. However, the study participants of this study reported that these facilities and equipment did not appear in enough quality and quantity. It was clear from the first author’s observation that most of the program sites included in this study had playing courts, which did not have leveled areas that are free from any obstruction. 

Although there has been a national initiative for sports-development programs across thirteen sporting events, the situation in the project sites does not back up that with evidence. Instead, implementing a youth volleyball development program in the sites studied, as some participants highlighted, compromises the quality of youth volleyball when we consider the limited facilities, resources, and materials present under existing conditions.

#### 3.2.2. Human Resource in the Sampled Projects 

In the sampled 10 youth volleyball project sites found in Amhara Regional State, the young volleyball players were school-age children found in selected high schools. In each high school, student–athlete samples were selected by using a stratified sampling method with the strata being gender. The questionnaire survey was distributed to 167 student–athletes from the 10 high schools studied, and all of them responded to it. Table 2 presents the demographic information of the study participants involved in filling out the open-ended items of the questionnaire.

As shown in Table 2, the volleyball players enrolled in the program were males (n = 120, 72%), and the rest, females (n = 47, 28%). In terms of schooling, 53 (32%) of the participants involved in the samples were in urban high schools in the three cities, while n = 114 (68%) were in rural high schools found in seven zones.

The youth volleyball players have been selected to participate in the youth volleyball program in their respective high schools based on their interest and previous experience in playing volleyball at the lower grade levels. As the FGD participant coaches stated, the assigned coach (PE teacher) in each high school hosting the volleyball program has the knowledge of the youth volleyball players’ experience, so selection or screening them was not difficult.

#### 3.2.3. Family Support

There were several problems/challenges identified by the sampled participant youth players, coaches, and program administrators. These problems/challenges are primarily family-related problems: lack of facilities and resources for the training, lack of attention by the program coordinators, and others. Here, it is important to note that almost all the study participants also believed that they faced several problems, particularly the absence of follow-up and support and the lack of training facilities and equipment.

#### 3.2.4. Media Support

Another concern involved the fact that sports media did not have a focus of attention on volleyball, particularly youth volleyball. Most of the interviewed participants and FGD discussants commonly agreed that sports media is an important tool for communication and creating awareness among the community members, and it needed to include programs addressing youth volleyball both at the regional and federal levels. The region also identified other issues such as coaches’ salaries, trainers and trainers’ sports equipment, various materials needed for the training, proper selection of training sites, and above all, raising awareness of the importance of the program for youth families and the local community.

### 3.3. Process Evaluation

Process evaluation includes mainly examining the implementation of a sport-based youth development program. Sports development often supports the assumption that young athletes’ multidimensional development may vary depending on how their sport training activities are designed and implemented, the extent of their participation or engagement in the youth sport development process, and the kind of positive relationship established between the youth volleyball players and their coaches. Figure 3 illustrates the core ingredients of the process evaluation that emerged from the data. 

As can be seen in Figure 3, the heart of the matter in the overall implementation evaluation is the youth volleyball development program. This program process is decomposed into the training, participation, and relationship components based on the evidence generated in this study. 

#### 3.3.1. Participation in Youth Volleyball

In all sampled project sites, youth volleyball players usually received 2 h of volleyball training per day and a total of 6 h per week. The weekly program was scheduled 3 times per week, including Monday, Wednesday, and Friday (afternoon, after school). The assigned youth coaches were selected on those days for training based on convenience to reduce the number of days students went to school for learning and volleyball program activities. The volleyball program activity made the training more suitable for the youth volleyball players because it did not clash with their study time so that their formal education remains unaffected due to the volleyball training. In terms of training hours, each session usually took two hours with breaks in between activities. 

However, most of the coach FGD discussants and some youth volleyball players perceived that these expectations were not strictly followed as there were missed training programs for several reasons related to lack of coaches’ interest and unexpected meeting schedules, large numbers of absentee youth players, and others. 

It was noted that when the program is mainly under the ownership of the office of education, management of these issues is relatively easier. For example, monitoring youth volleyball players who may miss the scheduled program may not be a problem. Not understanding that the number of shortcuts students take to avoid training will decrease during the project at school (Interviewee 1).

The coach participants in this study pointed out that the rate of attendance of youth volleyball players was not as expected. Most of them reported that attrition was a serious issue. According to one of the key interviewees (Interviewee 2), the regional state has taken measures to maintain the sustainability of the youth volleyball program throughout the region. These include developing and implementing a strategic plan to ensure that U-13 to U-15 to U-17 players remain constant. Hence, there is a holistic view of the youth volleyball talent development plan from the beginning to the end.

#### 3.3.2. Training at the Youth Volleyball Project Sites

As the FGD participants coaches reported, there were different components, including physical fitness, technical skills, tactical skills, and sporting behavior. In terms of each training session, there were warm-up activities such as general conditioning aerobics and some stretching exercises, followed by the main work, comprising the technical and tactical skills needed, and concluding activities, such as jogging and upper body stretching exercises. Almost all the FGD participant coaches agreed that their training mostly focused on physical fitness and technical skills more than the other activities. 

In regard to their coaching approach, the responses collected from the youth volleyball players’ open-ended questions showed a prevalence of a coach-centered approach to coaching in youth volleyball training settings. Empirical evidence in the literature in this field supports this claim [33]. Based on this, it is possible that the youth coaches need coach-development training to help them increase their coaching skills.

We also explored youth volleyball players and coaches’ perceptions of their experiences in the sampled volleyball project sites at Amhara Regional State. As per the evidence collected for this study, the sampled youth players believed that their experiences in the volleyball program provided them with many benefits, including the development of life skills (e.g., time management, communicating effectively, and sharing responsibility). 

In addition, these youth volleyball players believed that through their participation in the volleyball program, they also learned important values, such as teamwork, perseverance, honesty, and respect. In support of these, almost all the FGD participant coaches indicated that the young players’ participation in the volleyball projects was positive for their development, although many stated several challenges that hinder the attainment of far better outcomes. Some even believed that the players’ experience in the volleyball program was a wise use of their leisure time and contributed very positively toward their personal development. 

However, youth volleyball participants reported that their training was not measured by competition. In fact, “it is a good opportunity for almost all young people to participate in the competition, because the competition is part of the youth development program” (Coach Participant 3). Hence, conducting regular competitions between project sites at different times and creating a budget to recruit the best ones will benefit the region and the country. Furthermore, some coach participants suggest that the regional volleyball federation as well as the Federal Volleyball Federation should strengthen its monitoring and support schemes. 

Seen from the program coordinator’s side, the high school principals involved in FGD indicated that they were supportive of the practice of youth volleyball programs in their respective high schools, and they also recognized their importance for positive youth development. Furthermore, the principals mentioned that they encourage participation for all students and work to create a sporting environment that is in accordance with the program mission. However, this was not supported by the program implementers; that is, youth volleyball players and their coaches, who strongly argued that the follow-up and support of the program coordinators both at the high school as well as the woreda, zone, and regional levels were unsatisfactory. 

#### 3.3.3. Relationships of Coach and Athlete

Youth volleyball players and their coaches involved in this study reported that they had a positive relationship between them. Most of the youth players agreed that their coaches understood them and are willing to help. However, as some participant youths commented, their coaches did not have the commitment and proximity to them in matters related to training and development. 

### 3.4. Product Evaluation

#### 3.4.1. Effectiveness of the Program

Though a clear evidence file was not documented due to lack of data at the regional, zonal, and district/woreda levels, the sports office experts reported that the program contributed far less than was expected by the program planners. As they pointed out, those who transferred into clubs and those who joined the sports academies in volleyball sports each year are very few in number. As one FGD participant coach stated currently, the program does not seem to matter for those who are in charge of its implementation. In addition, another regional sports office expert commented that a strong monitoring and evaluation mechanism should be in place for better quality implementation of the program. 

As some study participants noted, the youth volleyball program appears to be a cost-effective investment of an increased use of local resources for positive youth development purposes, particularly in densely populated areas and in remote locations where few volleyball playing facilities exist. 

Overall, the youth volleyball program studied was effective in terms of an athlete-centered focus that emphasizes the positive youth development of each individual player. However, in terms of practice, study participants reported several problems that hinder the success of the program to produce its expected results.

#### 3.4.2. Impacts on the Development of Volleyball in Ethiopia

The relevance of the youth volleyball program for the development of volleyball in the regional state and country at large is undisputed. The different study participants commonly agreed that youth volleyball is important for the individual youth athlete, the region, and the country. In particular, the youth athlete participants viewed their participation in the volleyball program as a means of developing their sports performance. Most of them witnessed that participation in the volleyball program helped them to be healthy. Some study participants pointed to the relevance of the youth volleyball program for the development of talented youth volleyball players and improvement of the community attitude toward sports. Coaches also confirmed this fact. As most of them noted, a large number of young people came to participate in youth volleyball in their respective training sites. 

## 4. Discussion

Sports have the potential to promote multiple outcomes in youths’ development. For example, sports participation contributes to the holistic development of youth [34,35], surpassing the physical contribution to include other benefits, such as behavioral learning, character formation, and life-skills development. In the literature, sports development often supports the assumption that the developmental outcomes of sports participation also include other psychosocial outcomes that youth should learn to become productive citizens [36].

A sports policy consists of a set of materials for guiding the implementation of sports for development programs, procedures, intellectual dispositions, and ways of reasoning [37]. Indeed, the policy is at the heart of the strategies to guide success in youth development [38]. In terms of policy, many sports systems globally, including the Ethiopian national sports system, promote both excellence and participation [39]. Indeed, mass participation is the foundation for the advancement of sports excellence; hence their dual promotion is highly favored. Despite swinging between these intentions, many elite youth sports excellences typically come ahead of the participation. In this regard, few countries attain a balance between policies and resources that maximize the developmental benefits of youth sports [40]. However, the findings of this study indicate that, there was an apparent mismatch between the policy intention and the essential inputs. 

The talent-development environment refers to aspects of the physical, social, emotional, and cultural milieu, where athletic potentials are developed [41]. A holistic ecological perspective on talent development highlights the central role of the overall environment affecting the developmental trajectory of youth talent development perspective [42].

Family-related problems involved parents’ lack of awareness and lack of support, leading to disagreements with parents. The positive role of family during the youth athletes’ developmental years has been highlighted in the literature. For example, studies on elite youth athletes indicated that social support rendered by family, coaches, other athletes/peers, and support staff has a positive influence on players’ talent development. The finding of this study concerning the role of the family in the development of volleyball players’ talent is consistent with the findings reported in the literature on this field [43].

These problems are similar to the problems stated in other youth sports studies reported in Ethiopia. For example, a study reported similar problems faced by the youth athletes and their coaches in the sport of taekwondo in Addis Ababa, Ethiopia [44], in youth sport academies in Ethiopia [45,46], and in other similar regional youth sport programs as well [46]. Other problems included the busy schedule of coaches, lack of materials and facilities, considerable distance from home to project site (school), and lack of energy compensation, among others.

Youth volleyball players, including the sample participants in the current study, spend many training hours under the guidance of youth volleyball coaches in deliberate practice [47]. The hours spent training with coaches not only prepare youth volleyball players to develop skills to compete at junior elite levels but also to advance to senior elite levels and obtain life skills [48]. Hence, sports researchers and the wider sports community need to have a clear vision of the inherent value of youth sports participation. 

Three process features are identified by the study participants as important explanations of the youth volleyball development program, including opportunities to belong, positive social norms, and supportive relationships, among others [49]. It is also important that a suitable training environment, opportunities for broader physical, personal, and social skill development, and the presence of supportive interactions [50] be provided.

The interpersonal dynamics between the coach and the youth athlete are central to the youth athlete coaching process in any sport [51]. The interpersonal dynamics involved between coaches and their athletes have attracted researchers’ interest in the past. This is particularly true in youth volleyball sports as this relationship is critical for the success of the youth athletes’ development [52].

Among the various intentions stated in the strategic planning document, both at the national and regional state levels, the youth sports program anticipates attaining not only sporting excellence but also positive youth behavior outcomes [31]. The youth volleyball program studied was effective in terms of an athlete-centered focus that emphasizes the positive youth development of each individual player. However, in terms of practice, participants had several problems that hinder the success of the program to produce its expected results. In addition, the participant coaches, athletes, and administrators believed that youth volleyball is a context in which young athletes can learn not only sport specific talents but also life skills and values that facilitate PYD. There are numerous studies that support the positive benefits young players derive from participating in youth sporting programs [12,34,53]. 

Youth development outcomes are likely to occur when youths’ strengths are aligned with and supported by the resources of the environments in which they live [2]. The alignment of youths’ strengths and contextual resources is optimized when the youths participate in sports training activities that facilitate meaningful learning and sustained coach–athlete relationships [54]. Furthermore, it is optimized when the sports context presents opportunities for learning and development [55]. However, it should be noted that for the learning of sports skills, the youth need to practice in a sports setting. It is important to pay attention to how the sports environment is conducive to the learning of sports skills, positive behaviors, and life skills.

To continue enhancing the level of play in our volleyball, we need to develop more multi-skilled young volleyball players. This is only possible, if we are focusing on player development as opposed to team success in terms of winning [56]. Scholars argue that nothing may be beneficial if the team focuses its success exclusively on winning while undermining the youth volleyball players [57]. For example, the team might achieve successful results in competition, but this may be at the emotional expense and loss of developmental opportunity of the young players. 

Overall, youth volleyball is one of the relevant mechanisms to promote youth development processes and outcomes. However, the quality of the program components and the contextual factors associated with the program implementation, and the collective efforts of various stakeholder groups needs special attention [58]. The youth volleyball program studied and the associated contextual environments are often faced with challenges. This is also true in other sports [59], which may hinder not only sports talent development but also psychosocial development [60]. The findings in the current study provide evidence of attaining developmental outcomes within the Ethiopian youth sports context, in terms of program effectiveness and positive impacts, which are relevant to both the youth players and the sport itself. However, there are still challenges that should be seriously mitigated for further positive outcomes.

## 5. Conclusions and Implications

This study aimed to holistically explore the youth development program at Amhara Regional State, Ethiopia, by using the CIPP model. Based on the analysis of the collected data, it becomes clear that the youth volleyball program has several benefits for the players in increasing physical activity and health, positive interpersonal relationships, and emotional, cognitive, and behavioral competencies. The results of this study also indicate the presence of several challenges, including a lack of the necessary facilities and resources, lack of concern and settings, poor implementation practices, and minimal outcomes. In addition, the results indicate that the challenges of youth volleyball development in Ethiopia are complex and spread across the context, input, process, and products. Hence, when addressing the issues of youth volleyball, it is necessary to develop systems, processes, methods, and tools that recognize all these concerns. For young volleyball players to train smarter and perform better, coaches and program organizers need to understand the developmental needs of youth volleyball players and have a strategic intervention to empower them.

This study is helpful to increase insights about the youth volleyball development program from a broader perspective. Together, the findings of this study provide support for the value and contribution of youth volleyball programs and highlight several key areas for program designers to consider in future designs. For example, it is critical to prepare youth volleyball coaches through professional development and further training opportunities. Additionally, program designers should consider strengthening families’ participation in youth volleyball development programs.

## 6. Study Limitations and Possible Future Studies

This study was not free from limitations. Given that this study used a qualitative methodology, the authors did not establish a rigorous causal relationship between the different evaluation dimensions. Creating a transtheoretical quantitative model comprising variables representing the four domains of the CIPP model would be an important area of future research. One of the limitations of this study is that the sample included only youths aged 15 to 17. Future research should examine sport-based development programs across the other developmental periods (e.g., kids and seniors), as different age groups may be affected differently by the sports programs. Another potential limitation of this study is the absence of data collected from parents/guardians about youth volleyball development in the sports context. However, many studies show the significance of parental/guardian support for the effectiveness and impactfulness of a youth sports program. Hence, future research should include parental/guardians’ perceptions in studying a youth sports development program. Another limitation of this study is that it focuses exclusively on the effects of the youth development program within the sports setting leaving aside the school setting, thus ignoring the potential influences of other school-based programs and relationships. Future research may address this interconnection between the sports and school contexts.

## Figures and Tables

**Figure 1 healthcare-10-00719-f001:**
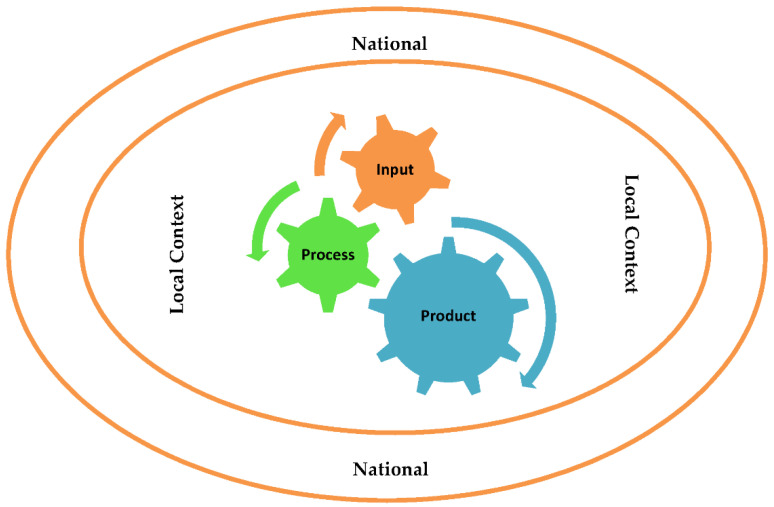
An innovative framework for evaluating youth volleyball programs. Source: Stufflebeam, D., Madaus, G. F., & Kellaghan, T. (2000) *Evaluation models* [22].

**Figure 2 healthcare-10-00719-f002:**
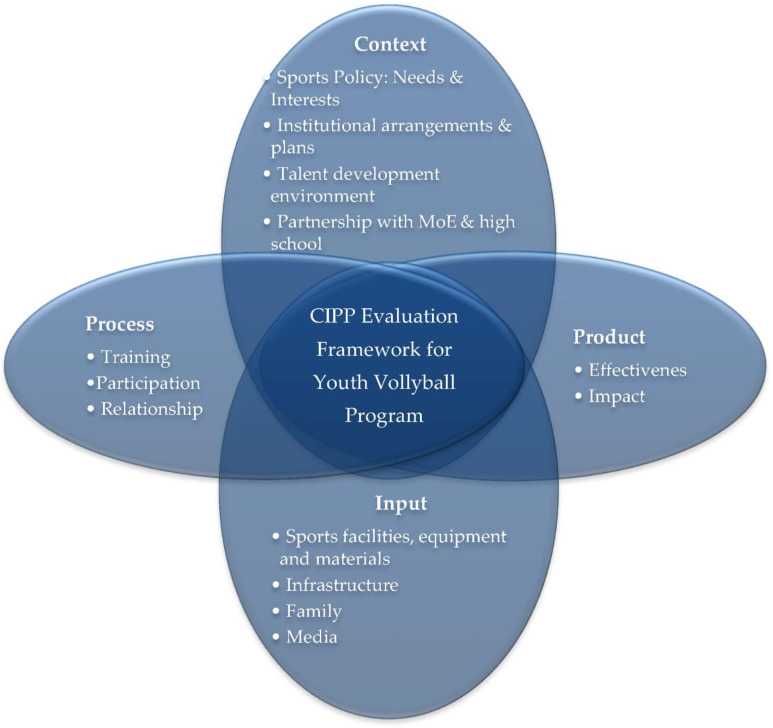
Hierarchical themes organizing framework.

**Figure 3 healthcare-10-00719-f003:**
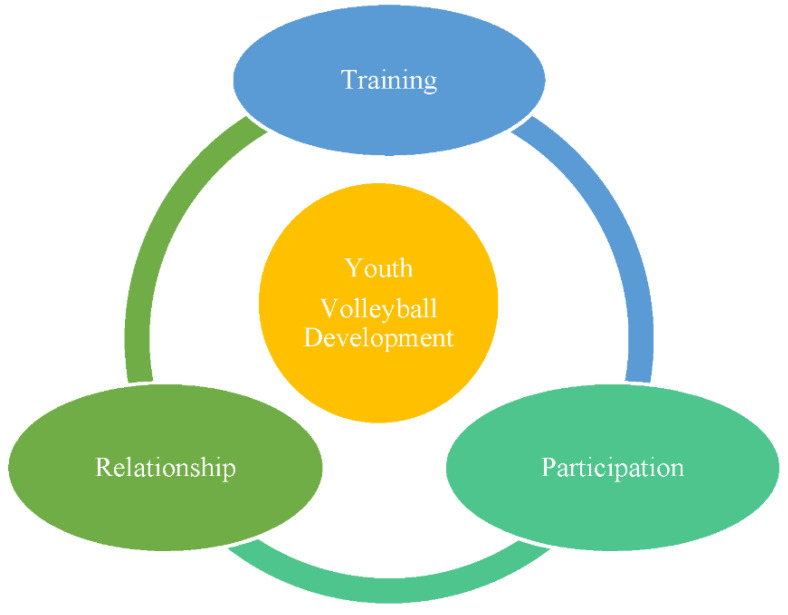
Process evaluation components.

**Table 1 healthcare-10-00719-t001:** Planned youth volleyball training schedule at Amhara Regional State, Ethiopia.

Program Duration	Daily	Weekly	Monthly
Dates	-	3	12
		Total	120 Days
Hours	2	6	24
		Total	240 h

Note: The overall program duration is ten months, lasting from September to June each year.

**Table 2 healthcare-10-00719-t002:** The demographics of the study participants involved in filling out the open-ended items of the questionnaire.

Characteristics of the Participants		Frequency	Percentage
Gender	Male	120	71.9
	Female	47	28.1
Age in years	14.00	1	0.6
	15.00	42	25.1
	16.00	91	54.5
	17.00	33	19.8
Education level	5 up to 8 grade level	58	34.7
	9 up to 10 grade level	86	51.5
	11 up to 12 grade level	20	12.0
	Other education levels	3	1.8
Total		167	100.0

## Data Availability

Not applicable.

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
