# Peer review of "Evaluating the Practices and Challenges of Youth Volleyball Development in Amhara Regional State, Ethiopia by Using the CIPP Model"

_healthcare, 2022, doi:10.3390/healthcare10040719_

Round 1

Reviewer 1 Report

I have one comment to do:

The conclusions must attend the study aim. Its confused.

Congratulations by excellent research.

Author Response

Dear Reviewer 1, 

Thank you so much for reviewing our manuscript and giving us valuable comments!

We have attached a file about our responses to your comments.

Thank you very much!

Reviewer 2 Report

The manuscript is interesting and it is presenting a topic, in a Country, that require attention. It is methodologically valid, but important revisions are required in the introduction and discussion. Furthermore, I strongly suggest to review also the language, not always it is clear and fluid.

The introduction is not clear and it is not presenting the topic properly. Some sentences are not connected each other, and according my opinion, all the introduction require an important revision.

The method, according my opinion, require a presentation of the questionnaire adopted (perhaps, the authors could add it in the supplementary materials). This could help the reader to better understand what the Authors asked to the sample.

Results and discussion together do not help the reader to appreciate what the Authors did. This section is long and complex, I think that the add of tables with few but essential and clear information about the findings, valorize the Authors’ work. Finally, I think that information about age, gender, height, weight of the sample studied (players and coaches) have to be added. I also suggest to include a section about possible future studies.

Thank you for your consideration

Author Response

Dear Reviewer 2,

Thank you so much for your valuable comments on our manuscript!

Here attached, please find a copy of our responses to your comments.

Kind regards!

Reviewer 3 Report

The article “Evaluating the practices and challenges of youth volleyball development in Amhara Regional State, Ethiopia using the CIPP model” submitted to MDPI group’s healthcare, analysed and described several youth volleyball development programs in Amhara Regional State, Ethiopia, through Daniel Stufflebeam’s Context, Input, Process, Product (CIPP) model. The paper is almost entirely descriptive, figures shown are limited in number and focused only on the design of the model. The reading of the article is difficult for the presence of several ill-formed sentences and the frequent jumping from one topic to another. In particular it has been difficult to follow the results and conclusions obtained by the authors without the support of tables and figures, which are usually shown in articles that use the CIPP model (Moghadas-Dastjerdi, T., Omid, A., & Yamani, N. (2020). Evaluation of health experts' education program for becoming multiprofessionals (family health caregiver) regarding health system transformation plan: An application of CIPP model. Journal of education and health promotion9, 227. https://doi.org/10.4103/jehp.jehp_75_20; Bilan, N., Negahdari, R., Hazrati, H., & Foroughi Moghaddam, S. (2021). Examining the quality of the competency-based evaluation program for dentistry based on the CIPP model: A mixed-method study. Journal of dental research, dental clinics, dental prospects15(3), 203–209. https://doi.org/10.34172/joddd.2021.034)

In my opinion the article must be improved in text and form so to be able to present the data obtained by the authors in a clearer and more detailed way.

Author Response

Dear Reviewer 3,

Thank you so much for reviewing our manuscript and giving us valuable comments!

We have attached a file about our responses to your comments.

Thank you very much!

Round 2

Reviewer 2 Report

Thank you to the Authors for the important improvements done but further work is required before I can suggest it for publication.

The introduction has been improved, thank you.

About the method section, from line 94 to line 99: I don’t think this section is appropriate for the methodology of the study. Furthermore, citation 17 and 18 are not appropriate for the study sector. I suggest to remove all this part.

I suggest to include the pre-planned interview guide in the supplementary material. I also suggest to add in the supplementary material the two open-ended items

Results

From line 260 to 267 I suggest to move this, according to me it is methods not results

Figure 2 is not clear. In the context, I cannot read the second part.

Line 300: according to my opinion, insert objective in the results section create confusion. I strongly suggest to focalize the results only on the finding of the study

Please, improve table 2. After age indicate that it is in years.

Please add the conclusions.

The English has been improved but the manuscript is too long. The results section is too narrative, according my opinion, this section of the manuscript should have to be synthetized as much as possible.

Author Response

Dear Reviewer 2,

Thank you so much for your critical comments!

Here attached, please find our responses to your comments.

Again, thank you so much for your support.

Reviewer 3 Report

Dear authors,

I'm glad to see a good improvement in the drafting of the paper. The text is now much clearer and more fluid.

I also understand your decision to split the data into a qualitative article and quantitative article. 

I just make a few little notes:

1) Text in Figure 1 should be alined;

2) In Figure 2 text needs to be adjusted because is partially covered; text font in red could be changed in other color font (perhaps white);

2) line 686 there is a "please add:" to remove

Author Response

Dear Reviewer 3,

Thank you so much for your critical comments!

Here attached, please find our responses to your comments.

Again, thank you so much for your support!
